# Learning Chern Numbers of Multiband Topological Insulators with Gauge Equivariant Neural Networks

**Longde Huang**
Department of Mathematical Sciences
Chalmers University of Technology
University of Gothenburg
`longde@chalmers.se`

**Oleksandr Balabanov**
Department of Physics
Stockholm University

**Hampus Linander**
VERSES AI
Los Angeles, CA, USA
`hampus.linander@verses.ai`

**Mats Granath**
Department of Physics
University of Gothenburg
`mats.granath@physics.gu.se`

**Daniel Persson**
Department of Mathematical Sciences
Chalmers University of Technology
University of Gothenburg
`daniel.persson@chalmers.se`

**Jan E. Gerken**
Department of Mathematical Sciences
Chalmers University of Technology
University of Gothenburg
`gerken@chalmers.se`

## Abstract

Equivariant network architectures are a well-established tool for predicting invariant or equivariant quantities. However, almost all learning problems considered in this context feature a global symmetry, i.e. each point of the underlying space is transformed with the same group element, as opposed to a local *gauge* symmetry, where each point is transformed with a different group element, exponentially enlarging the size of the symmetry group. We use gauge equivariant networks to predict topological invariants (Chern numbers) of multiband topological insulators for the first time. The gauge symmetry of the network guarantees that the predicted quantity is a topological invariant. A major technical challenge is that the relevant gauge equivariant networks are plagued by instabilities in their training, severely limiting their usefulness. In particular, for larger gauge groups the instabilities make training impossible. We resolve this problem by introducing a novel gauge equivariant normalization layer which stabilizes the training. Furthermore, we prove a universal approximation theorem for our model. We train on samples with trivial Chern number only but show that our model generalizes to samples with non-trivial Chern number and provide various ablations of our setup.

## 1 Introduction

Geometric deep learning is a subfield of machine learning that takes advantage of the geometric and topological structures inherent in complex data to construct more efficient neural network architectures [1]. This approach has been successfully applied in a variety of domains, from medical imaging [2] to high-energy physics [3] and quantum chemistry [4]. As we show in this paper, this perspective is particularly valuable for studying topological insulators, a class of materials which has been one of the main areas of interest in condensed matter physics over the last two decades [5, 6],

39th Conference on Neural Information Processing Systems (NeurIPS 2025).

with a broad range of applications, including spintronics and magnetoelectronics [7], photonics [8], quantum devices [9], and quantum computing [10, 11, 12].

The mathematical field of topology studies objects which cannot be deformed continuously into each other. For instance, a doughnut is topologically equivalent to a coffee cup (the hole in the doughnut becoming the hole in the handle) but not to a ball (which does not have a hole). Topological insulators are materials whose interior is insulating yet whose surface or boundary is conducting, with a prime example being materials characterized by a *Chern number*. The Chern number is an integer topological quantity that originates from the fact that the phase of a quantum mechanical wave-function is not a physically measurable quantity, thus corresponding to a *gauge symmetry*. The fact that a material with finite Chern number cannot be continuously deformed into a material with Chern number 0 without closing the energy gap (which make the materials insulating) implies that there has to be a zero energy (conducting) state at the boundary.

So far, deep learning models have only been able to predict Chern numbers for materials with the abelian gauge group $\mathrm{U}(1)$, corresponding to 1 filled band [13, 14]. However, these models fail to learn Chern numbers for higher-rank gauge groups $\mathrm{U}(N)$, pointing to a fundamental challenge in the multi-band regime $N > 1$.

We identify the gauge symmetry of the system as the central reason for the failure of traditional approaches in the high-band setting and instead propose to use a gauge equivariant network for learning topological invariants such as the Chern number. Consider a neural network $\mathcal{N} : X \to Y$ which maps a set of inputs $x = (x_1, \ldots, x_n) \in X$ (one for each site in a lattice) into a set of outputs $y = \mathcal{N}(x_1, \ldots, x_n) = (y_1, \ldots, y_n)^\top \in Y$. Usual group equivariant networks $\mathcal{N}$ satisfy the constraint $\mathcal{N}(\rho_X(g)x_1, \ldots, \rho_X(g)x_n) = (\rho_Y(g)y_1, \ldots, \rho_Y(g)y_n)^\top \ \forall g \in G$ with symmetry group $G$ and representations $\rho_{X,Y}$ on the input- and output spaces, respectively. In contrast, a network is *gauge equivariant* if it is equivariant under the action of different elements of the symmetry group $G$ on the different inputs, i.e. $\mathcal{N}(\rho_X(g_1)x_1, \ldots, \rho_X(g_n)x_n) = (\rho_Y(g_1)y_1, \ldots, \rho_Y(g_n)y_n)^\top$.

In a discretized input domain $X$ in $d$ dimensions with $p$ points per dimension, a gauge symmetry effectively means that the total symmetry group of the problem consists of $p^d$ copies of $G$. This exponential enlargement of the symmetry group explains why non-equivariant networks are often unable to learn tasks which feature a gauge symmetry. We therefore claim that gauge-equivariant networks are necessary to learn high-band Chern numbers of topological insulators. This is a novel application for gauge equivariant neural networks, which thus far have mainly found applications within the realm of lattice gauge theories for quantum chromodynamics. In particular, we cast the problem at hand in a form in which we can use an adapted version of the Lattice Gauge Equivariant Convolutional Neural Networks (LGE-CNNs) [15] to learn multiband Chern numbers.

However, LGE-CNNs have been plagued with instability problems. These networks feature bilinear layers which frequently lead to exploding or vanishing gradients, making the training process unstable. With growing depth or for larger gauge groups, this problem becomes more severe, limiting the applicability of LGE-CNNs. We provide a solution by introducing a new gauge equivariant normalization layer that stabilizes the training and leads to consistent convergence of deep networks.

Our main contributions are as follows:

- We resolve the instability issues of LGE-CNNs by introducing a new gauge equivariant normalization layer. By training our purely local network with a novel combinations of loss functions, we obtain a model which generalizes from trivial to non-trivial Chern numbers and to unseen lattice sizes.

- We prove a universal approximation theorem for our architecture in this context which shows that our model can approximate all $\mathrm{U}(N)$ gauge invariants arbitrarily well. We perform ablations over different gauge equivariant architectures motivated by this theoretical investigation.

- We provide a novel application of $\mathrm{U}(N)$-gauge equivariant neural networks to the task of learning higher-band Chern numbers. The resulting model can predict Chern numbers in two dimensions for systems with at least $N = 7$ filled bands. In contrast, previous models could only handle the trivial case of a $\mathrm{U}(1)$-symmetry.

- Our model is also the first to be able to learn higher-dimensional ($D = 4$) Chern numbers of multi-band topological insulators.

## 2 Literature Review

Equivariance under global matrix groups have been considered in [16] and for the orthogonal groups in [17]. Gauge equivariant networks have been considered in two different settings. In the first setting, the gauge symmetry concerns local coordinate changes in the domain of the feature maps [18, 19, 1]. This case was first studied theoretically in [20] and models respecting this symmetry were introduced in [21, 22]. Applications of gauge equivariant networks to lattice quantum chromodynamics (QCD) fall into the second setting, where the gauge transformations act on the co-domain of the feature maps. An important problem in lattice QCD is sampling configurations from the lattice action, a problem for which gauge equivariant normalizing flows [23, 24, 25, 26, 27] as well as gauge equivariant neural-network quantum states [28] have been used. In contrast, our model is based on a gauge equivariant prediction model developed for lattice QCD [15].

Machine learning for quantum physics has seen an explosive development over the last decade with applications in condensed matter physics, materials science, quantum information and quantum computing to name a few [29, 30, 31, 32]. In this brief overview we focus on deep learning and applications to topological states of matter. Early ground-breaking work in this area includes [33] that used supervised learning on small convolutional neural networks for identifying the ground state of the Ising lattice gauge theory, as well as [34] that developed an unsupervised method "learning by confusion" to study phase transitions including topological order of the Kitaev chain. Of particular relevance to our work are the papers [35, 13] that used convolutional neural networks and supervised learning to predict U(1) topological invariants. This work was later extended to an unsupervised setting in [14, 36] by incorporating the scheme of learning by confusion and augmenting data using topology-preserving deformations.

## 3 Learning Multiband Chern Numbers

In this section, we will introduce the learning task of predicting multiband Chern numbers. We will first outline the features, targets, and relevant symmetries and then demonstrate that two naive approaches of learning in this context fail. The discussion of how the learning task presented here is related to the physics of multiband topological insulators is relegated to Appendix A.

### 3.1 Features, targets and symmetries

For most of our experiments, we consider features on a two-dimensional rectangular lattice $\Lambda$ with periodic boundary conditions with a total of $N_x \times N_y = N_{\text{site}}$ grid points. At each grid point $(i, j)$, we have features which are complex unitary $N \times N$ matrices $W_{i,j} \in \mathrm{U}(N)$, the so-called Wilson loops. These will be the inputs for most of our models. The Wilson loops can be written in terms of Hermitian link matrices $U_{i,j}^x, U_{i,j}^y \in \mathbb{C}^{N \times N}$ which are attached to the edges in $x$- and $y$ directions connected to the grid point $(i, j)$. The Wilson loops are given by a product of links along a $1 \times 1$ loop of edges,

$$W_{i,j} = U_{i,j}^x U_{i-1,j}^y U_{i-1,j-1}^x U_{i,j-1}^y. \tag{1}$$

The learning target is the Chern number $\tilde{C}$ defined by

$$\tilde{C}\big(\{W_{i,j} \mid (i,j) \in \Lambda\}\big) = \sum_{(i,j) \in \Lambda} \operatorname{Im} \operatorname{Tr} \log W_{i,j}. \tag{2}$$

It can be shown that $\tilde{C}$ defined in this way is an integer [37]. The learning task we want to study in this article is to predict the Chern number $\tilde{C} \in \mathbb{Z}$ given the Wilson loops $\{W_{i,j} \mid (i,j) \in \Lambda\}$. Although the Chern number is given by the innocuous-looking equation (2), learning it is not straightforward as demonstrated in the next subsections. We will show that the main reason for the difficulty of learning (2) lies in the gauge invariance of $\tilde{C}$.

Due to the gauge invariance in the mathematical description of topological insulators, the Chern number $\tilde{C}$ is invariant under transformations $W'_{i,j} = \Omega_{i,j}^\dagger W_{i,j} \Omega_{i,j}$ of the Wilson loops, where crucially the transformation matrices $\Omega_{i,j} \in \mathrm{U}(N)$ depend on the lattice site. The fact that the symmetry holds even if the Wilson loops at different lattice sites are transformed with different

group elements of U($N$) makes this into a gauge symmetry which is exponentially large in $N_\text{site}$. In summary, $\tilde{C}$ satisfies

$$\tilde{C}\big(\{W_{i,j} \,|\, (i,j) \in \Lambda\}\big) = \tilde{C}\big(\{\Omega_{i,j}^\dagger W_{i,j} \Omega_{i,j} \,|\, (i,j) \in \Lambda\}\big) \quad \forall \{\Omega_{i,j} \in \text{U}(N)\}. \tag{3}$$

In the following two sections, we will demonstrate that two naive approaches to learn (2) both fail, motivating the use of LGE-CNNs which are by construction equivariant with respect to local gauge transformations.

## 3.2 Learning Chern numbers using ResNets

Using the fact that $\text{Tr}\log(X) = \log\det(X)$, it follows that the Chern number defined in (2) can be written as a sum over $\text{Im}\log\det(W)$. As a warmup to predicting Chern numbers, we start by considering the simpler problem of predicting determinants of $N \times N$ real matrices $A$.

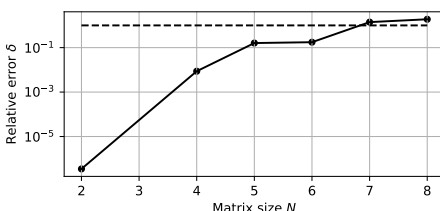

Figure 1: Best relative error of predicted matrix determinants for polynomial architectures as a function of increasing matrix size. The ablations include layers up to order 4. Dashed line indicates relative error of a mean predictor. Architectures considered include layers of order $\leq 4$, and depth $\leq 4$, containing terms of up to order 16 by composition.

Table 1: Relative error $\delta$ for linear MLP and bilinear residual architectures predicting the determinant of real $4 \times 4$ matrices with uniform random elements in $[0, 1]$. Standard MLP architecture fails to learn the determinant relation.

| Architecture | Layers | $\delta$ |
|---|---|---|
| MLP | 2 | 1.02 |
|  | 3 | 1.02 |
| Bilinear | 2 | 0.01 |
|  | 3 | 0.01 |

We construct a dataset containing $N \times N$ matrices with elements sampled from a uniform distribution on the unit interval $[0, 1]$. As a baseline, we use a naive multilayer perceptron (MLP) with residual connections $f : \mathbb{R}^{N^2} \to \mathbb{R}$, taking the matrix elements as input and predicting the determinant value.

The determinant of an $N \times N$ matrix can be expressed as an order $N$ polynomial in the matrix elements. Inspired by this, we consider higher order layers with structure

$$A_{ij}^\text{out} = \sum_{k_1, \ldots, k_{2R}} \theta_{ij}^{k_1 \ldots k_{2R}} A_{k_1 k_2}^\text{in} \ldots A_{k_{2R-1} k_{2R}}^\text{in}, \tag{4}$$

where $R$ is the number of factors of $A$ in the layer, and $\theta_{ij}^{k_1 \ldots k_{2R}}$ are learnable parameters. An architecture containing layers with $R = 2$ will be referred to as bilinear. As the number of parameters grows quickly with order $R$, we use layers of order $R \leq 4$ and construct higher order terms by composition of multiple layers and residual connections. Predicted determinants $f(A)$ are evaluated against the target determinant value $\det(A)$ using absolute relative error $\delta = |f(A) - \det(A)| \,/\, |\det(A)|$.

Table 1 shows the failure of a residual MLP architecture to learn the determinant of $4 \times 4$ real matrices with uniformly distributed elements on $[0, 1]$, whereas a bilinear architecture with layers of order 2 can achieve low relative error. Even though these higher order layers provide an architecture that is expressive enough for determinants in small dimensions, they quickly run into issues for larger matrices. Figure 1 shows that for matrix sizes corresponding to band size $\geq 4$, learning the determinant becomes prohibitively hard without better model priors. See Appendix F for more details on the architecture ablation.

## 3.3 DeepSpec: Baseline Equivariant Model

The gauge symmetry of the system implies the equivalence relation

$$W \sim \Omega^\dagger W \Omega, \quad \forall \Omega \in \text{U}(N) \tag{5}$$

on the set of Wilson loops at each grid point. According to the spectral theorem, there is exactly one diagonal matrix in each equivalence class (group orbit) with elements in U(1), up to reordering

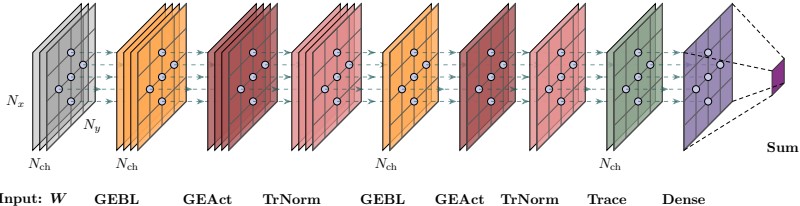

Figure 2: Architecture of GEBLNet. In this figure, the rectangles represent the spatial grid, and the number of layers ($N_{\text{ch}}$) represents the number of channels ($\gamma$). Each circle represents a site on the grid, and quantities on different sites do not interact with each other, until the last summation on grids.

of the diagonal elements. In other words, the set of equivalence classes for this relation is given by $U(1)^N/_{S_N}$. This fact has been used in the construction of gauge equivariant spectral flows [24]. Based on this, we construct a baseline invariant model DeepSpec.

DeepSpec preprocesses the flux data with a diagonalization, and then feeds the eigenvalues into a DeepSet-like model, which is invariant under permutation of the input data, calculating locally the output as $\mathcal{N}(\lambda_k^i)_k = \rho\left(\sum_i \phi(\lambda_k^i)\right)$ where $\rho,\ \phi$ are arbitrary neural networks, set as MLPs.

We train DeepSpec to predict Chern numbers on $5 \times 5$ grids while gradually increasing the number of bands. The data generation scheme is consistent with other experiments, as formally introduced in Section 6. The model successfully learns Chern numbers for the trivial case ($N = 1$), but fails as the number of bands increases and the system complicates. Specifically, the accuracy at evaluation drops drastically to $44.7\%$ for $N = 2$, and $7.8\%$ for $N = 4$.

Rather than enforcing group equivariance layer-by-layer, DeepSpec predicts gauge-invariant quantities by restricting its input to invariant features only. We attribute its failure to this structural characteristic and thus aim to develop models that manifestly respect gauge equivariance.

## 4 Network Architecture

As demonstrated in the previous section, even hand-crafted polynomial architectures fail to learn a simplified version of the Chern number. Furthermore, naive equivariant models that operate solely on invariant features, such as DeepSpec, also struggle to effectively learn the target quantity. Motivated by the performance of our gauge equivariant networks, we propose that this is due to the size of the gauge symmetry present in this problem. Since the Wilson loops at each site can be transformed independently, the total symmetry group is exponentially larger than the symmetry groups of more traditional group equivariant networks.

The input data in our network is the set of discretized Wilson loops $W_k^\gamma \in U(N)$ and all equivariant layers in our setup operator on tensors of this form. Here, we use the shorthand $k = (i, j)$ for the lattice point. The index $\gamma$ counts the number of different orientations of the Wilson loops per site (in 2D, there is only one) for the input and serves as a general channel index in deeper layers. Hence, our layers operate on complex tensors of the shape $N_{\text{ch}} \times N_{\text{sites}} \times N \times N$.

### 4.1 Gauge equivariant layers

Our model is composed of the following equivariant layers, which were introduced in [15] as well as our new gauge equivariant normalization layer.

**GEBL (Gauge Equivariant Bilinear Layers)**  Given an input tensor $W_k^\gamma$, the layer computes a local quantity per site as $W_k'^\gamma = \sum_{\mu,\nu} \alpha_{\gamma\mu\nu} W_k^\mu W_k^\nu$, where $W'$ has $N_{\text{out}}$ channels and $\alpha_{\gamma\mu\nu} \in \mathbb{C}^{N_{\text{in}} \times N_{\text{in}} \times N_{\text{out}}}$ are trainable parameters. This is the primary feature mixing layer, since it not only extends the feature space, and also captures higher order terms of the original flux data $W$, which is demonstrated to be crucial for the model's expressivity in Section 5. It can easily be checked that this layer is equivariant. In practice, GEBL includes also a linear and a bias term which are obtained by

enlarging $W$ with its Hermitian conjugate and the identity matrix. In order to merge two branches of the network, two different $W$ can also be used on the right-hand side.

**GEAct (Gauge Equivariant Activation Layers)**  Given a tensor $W_k^\gamma$, the layer maintains a channel size of $N_{\text{in}}$ and serves as an equivariant nonlinearity defined by $W_k'^\gamma = \sigma(\text{Tr}W_k^\gamma)W_k^\gamma$, where $\sigma$ is a usual activation function. In Section 5, we prove a universal approximation theorem for a certain type of $\sigma$. In practice, we use $\sigma(z) = \text{ReLU}(\text{Re}z)$ to avoid gradient vanishing; hence, we also refer to this layer as GEReLU.

**GEConv (Gauge Equivariant Convolution Layers)**  Given a tuple $(U_k^\mu, W_k^\gamma)$, the layer performs a convolution as

$$W_k'^\gamma = \sum_{\mu,d} \sum_\sigma \omega_{\mu d\gamma\sigma}(U_{k+d\hat{\mu}}^{d\mu})^\dagger W_{k+d\hat{\mu}}^\sigma U_k^{d\mu}, \tag{6}$$

where $U_k^{d\mu} = U_k^\mu \ldots U_{k+(d-1)\hat{\mu}}^\mu$ and $d\hat{\mu}$ are the length-$d$ vectors in the $\mu$ direction in the lattice. Mathematically, this layer is capable of simulating the integral of the flux over any closed area, therefore, it is the only layer in our setup that uses link variables $U_k^\gamma$ and introduces interactions between neighboring points.

The output $W'$ of this layer has the shape $N_{\text{out}} \times N_{\text{site}} \times N \times N$ and $\omega$ are trainable weights of shape $N_{\text{dim}} \times d \times N_{\text{out}} \times N_{\text{in}}$. Note that this layer does not update the links. Using the transformations (15) of the link variables and the transformation of the Wilson loops (3), one can verify that this layer is equivariant as well. When we take a zero convolution kernel size, the layer degenerates into an equivariant linear layer that is completely local.

**Trace Layer**  Given a tensor $W_k^\gamma$, this layer maintains the channel size and takes the trace of the fluxes as $T_k^\gamma = \text{Tr}W_k^\gamma$. Since the trace is invariant under the transformations (3), this layer renders the features gauge invariant. The output has the shape $N_{\text{in}} \times N_{\text{site}}$.

**Dense Layer**  After the trace layer, we perform a real valued linear layer on $T_k$ as $T_k' = w_{\text{Re}} \cdot \text{Re}(T_k) + w_{\text{Im}} \cdot \text{Im}(T_k) + b$ for the final prediction, where $w_{\text{Re}}$, $w_{\text{Im}}$ and $b$ are trainable parameters. The output features have shape $N_{\text{out}} \times N_{\text{site}}$. Note that we only transform gauge invariant features with this layer since it does not respect the gauge symmetry.

**TrNorm (Trace Normalization Layers)**  The bilinear GEBL-layers introduced above quickly lead to training instabilities when stacked deeply, as demonstrated in Section 6 below. To solve this problem, we introduce a novel gauge-equivariant normalization layer which we insert after the nonlinearities. Given an input tensor $W_k^\gamma$, this layer maintains the channel size and performs a channel-wise normalization as

$$W_k'^\gamma = \frac{1}{|\text{mean}_\gamma\{\text{Tr}W_k^\gamma\}|}W_k^\gamma. \tag{7}$$

This operation is gauge equivariant since the prefactor is gauge invariant. After the normalization, the output features $W'$ satisfy $\text{mean}_\gamma\{\text{Tr}W_k'^\gamma\} = e^{i\phi_k}$ for some $\phi_k \in \mathbb{R}$.

## 4.2  Network Architecture

We now use the layers introduced in the previous section to construct three different equivariant network architectures.

**GEBLNet (Gauge Equivariant Bilinear Network)**  GEBLNet is a model that only operates locally, i.e. the inputs are the fluxes $W_k^\gamma$ alone, and features at different sites only interact with each other in the final sum. It processes the fluxes through repeated blocks of GEBL , GEAct , and TrNorm layers. The outputs are then aggregated through a Trace layer, followed by a dense and a summation over sites to produce the prediction on the Chern number. This is our primary model to study. An example structure is shown in Figure 2.

**GEConvNet (Gauge Equivariant Convolutional Network)** GEConvNet is a model that features GEConv layers, therefore takes both the links ($U^\mu$) and the fluxes ($W^\gamma$) as input. Each GEBL-GEAct-TrNorm block, similar to that of GEBLNet, is paralleled by a GEConv-GEAct-TrNorm block, with their outputs combined through a subsequent GEBL layer. This process is repeated through several iterations, after which the resulting output is passed to the Trace- Dense- and sum sequence to produce the final prediction. An example structure is shown in Figure 8 in Appendix E.

## 5 Theoretical Foundations of the Model

In this section we will present a universal approximation theorem for our models. We focus on GEBLNet, whose inputs are solely $W_k$. Recalling the equivalence relation (3), the local quantity is in fact a class function on the gauge group $U(n)$, which is defined as follows:

**Definition 5.1.** A (complex) class function on a Lie group is a function $f : G \to \mathbb{C}$ such that $f(g^{-1}hg) = f(h)$, for all $g, h \in G$.

We denote the closed subspace formed with square integrable class functions as $L^2_{class}(G)$.

We now present our main theoretical result, which shows our model's capability to learn an arbitrary class function.

**Theorem 5.2** (Universal Approximation Theorem). *For a compact Lie group $G$, and with the nonlinearity $\sigma$ in GEAct taking the form $\tilde{\sigma} \circ Re$, where $\sigma$ is bounded and non-decreasing, GEBLNet could approximate any class function on $G$.*

The full proof is presented in Section D in the Appendix. Here we present a brief sketch of it.

*Sketch of Proof.* Consider a network with $M$ GEBL layers and sufficient width. One can show that the output of such a network can approximate any function of the form

$$w_i \sigma(\text{Re} \sum_{j=0}^{2^M} \alpha_{ij} \text{Tr} g^i)(\sum_{j=0}^{2^M} \alpha_{ij} \text{Tr} g^i) + b. \tag{8}$$

However, this is equivalent to a one-layer MLP on the polynomials $(\text{Tr} g, \dots, \text{Tr} g^{2^M})$. Therefore it can approximate any function in $\{f(\text{Tr} g, \dots, \text{Tr} g^{2^M})\} \cap L^2(G)$. By showing that the space $\bigcup_M \{f(\text{Tr} g, \dots, \text{Tr} g^M)\} \cap L^2(G)$ is dense in $L^2_{\text{class}}(G)$, we conclude that (8) can approximate any function in $L^2_{class}(G)$. □

The argument above can be generalized to the following theorem in higher dimensions if we restrict the group $G$ to be the unitary group, and we discuss this further in Section D in the Appendix.

**Theorem 5.3** (Higher-Dimensional UAT for GEBLNET). *Denote the unitary group as $G$. Under the same assumptions on the nonlinearity,* GEBLNET *can approximate arbitrarily well any square-integrable function $f \in L^2(G^K)$ such that*

$$f(g_1, \dots, g_K) = f(hg_1h^{-1}, \dots, hg_Kh^{-1}), \quad \forall h \in G. \tag{9}$$

## 6 Experiments

To assess the performance of our gauge-equivariant neural network, we conducted extensive numerical experiments. As is standard in the literature in this domain [13, 34, 35, 38, 39], we train on synthetically generated data. In particular, we train on uniformly distributed link variables and on link variables whose distribution was adjusted for a specific distribution of Chern numbers. These allow us to test our model on input configurations spanning a wide range of Chern numbers with many filled bands, going beyond the Hamiltonian models common in the literature. Our datasets have varying grid sizes and distributions and are based on $2D$ grids, unless otherwise stated. Data samples have the form $(U_k^x, U_k^y, W_k)_k$.

## 6.1 Data Generation

**General Dataset** The data generation pipeline consists of two main steps. Given a grid size $N_x \times N_y$, we first generate random link variables using the following algorithm:

1. For $\mu = x, y$, draw $A_k^\mu \sim \mathcal{N}(0,1)^{N \times N}$.
2. Perform a QR decomposition on $A_k^\mu$, decomposing it into the product of a unitary matrix $U_k^\mu$ and a semi-definite matrix $\Sigma_k^\mu$, $A_k^\mu = U_k^\mu \Sigma_k^\mu$.
3. Use $U_k^\mu$ as the link variable for site $k$ in direction $\mu$.

It can be shown that the distribution of the links generated in this way is uniform on $\mathrm{U}(N)$, i.e. the random variables $U$ and $gU$ are identically distributed for all $g \in \mathrm{U}(N)$. See Appendix C for details. In the next step, we compute the fluxes $W_k$ using (1) and the discrete Chern number $\tilde{C}$ using (2). Ultimately, this yields a dataset of data-value pairs $((U_k^\mu, W_k^\gamma), \tilde{C})$. We generate the training samples continuously during training to avoid overfitting.

**Diagonal Dataset** For some of our experiments, we require control over the distribution of the Chern numbers in our training data. To this end, we employ a different data generation strategy. Due to the invariance of our model under gauge transformations, training samples which lie in the same gauge orbit are equivalent in the sense that the parameter updates they induce are the same. This implies that we can select an arbitrary element along the gauge orbit for training. As discussed in Section 3.3, there is always a diagonal matrix with $\mathrm{U}(1)$-valued components in the orbit. Therefore, by training on these matrices in $\mathrm{U}(1)^N$ and manipulating the distribution of the diagonal values, we can generate datasets with different distributions of Chern numbers. Note that networks trained on these datasets of course generalize to non-diagonal Wilson loops since they are gauge invariant. Detailed discussions are provided in Appendix C.

## 6.2 Training and Evaluation

We adopt two loss functions for our training: the global loss $L_{\mathrm{g}}$ and the standard deviation loss $L_{\mathrm{std}}$. Specifically, given the network $f(W)$ and the Chern number $\tilde{C}$, $L_{\mathrm{g}}$ calculates $\|f(W) - \tilde{C}\|_1$. Meanwhile, $L_{\mathrm{std}}$ evaluates the entrywise standard deviation of the network output after the dense layer, denoted as $(g(W_k))_k$, namely $\|\max(\{\mathrm{std}(g(W_k)_k), \delta\}) - \delta\|_1$, where $\delta$ is a hyperparameter, set to 0.5 by default. The standard deviation loss is necessary to prevent the model from collapsing to only zero outputs, since it forces the model to output locally different quantities. This is particularly relevant for the training on trivial topologies only, as described below. The total loss function $L_{\mathrm{total}}$ adds these two terms, $L_{\mathrm{total}} = L_{\mathrm{g}} + L_{\mathrm{std}}$.

For evaluation, we compute the accuracy by rounding the network output $f(W)$ to the nearest integer and comparing it with the Chern number $\tilde{C}$, unless otherwise stated. For the main GEBLNet, we set a representative model configuration, whose GEBL layers and hyperparameters are listed in Table 4 in Appendix E.

## 6.3 Experimental Results

**Model Comparison** For a basic model comparison, we train GEBLNet, GEConvNet, and TrMLP on 2D grids to learn Chern numbers by training on non-diagonal data. We find that GEBLNet outperforms the other models in both accuracy and robustness, see Figure 9 in the Appendix.

Using a benchmark grid size of $5 \times 5$ and $N = 4$ filled bands, GEBLNet achieved approximately $95\%$ accuracy across different seeds, demonstrating strong robustness. In contrast, GEConvNet struggled to learn correct results with positive kernel sizes, likely due to redundant information. We also tested a degenerate GEConvNet with kernel size 0 (a local network) which performed better than its non-local counterpart, but it remained less robust than GEBLNet. A complexity-accuracy comparison in Figure 9 in Appendix C demonstrates the balance achieved by the representative model, which performed consistently well while maintaining efficiency. Additionally, tests on increasing band sizes (Table 2) show that the representative model effectively learns Chern numbers up to 7 bands, retaining high accuracy and robustness.

Table 2: Accuracy of GEBLNet trained and evaluated on a $5^2$ grid.

| Bands | 4 | 5 | 6 | 7 | 8 |
|-------|-----|-----|-----|-----|-----|
| Accuracy | 95.9% | 94.0% | 93.8% | 91.7% | 52.5% |

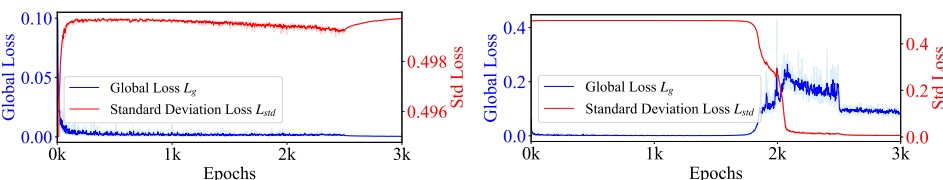

Figure 3: Comparison of global and standard deviation loss on validation data between the same two runs learning on only zero Chern numbers, as shown in Figure 4. The former, without TrNorm layers, collapses to zero local quantities everywhere, hence having a lower $L_g$ on trivial samples, yet could not generalize to nontrivial cases. In contrast, the latter, with TrNorm layers, succeeds in learning global quantities and local differences simultaneously.

**Training on Trivial Topologies** In order to test the generalization properties of our model, we train exclusively on topologically trivial samples. To achieve this, non-trivial samples were manually filtered out during data generation, turning $L_g$ effectively into $\|f(W)\|_1$.

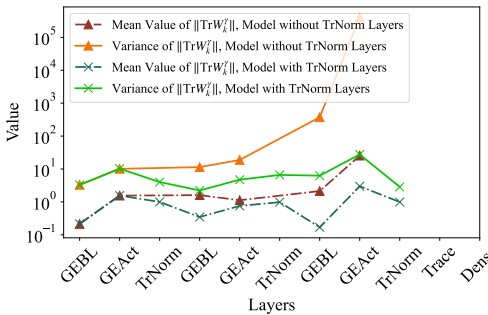

Figure 4: Comparison of statistics of $\|\mathrm{Tr}w_k'^\gamma\|$ across each layer, between two training runs on a $5 \times 5$ grid, with $4$ filled bands, with or without TrNorm Layers.

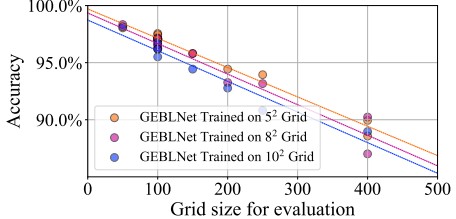

Figure 5: Comparison of model accuracy across different grid sizes. Each run, represented by markers of the same color, has identical configurations, but is trained on grids of a different size. Each line represents a linear regression on the corresponding run.

A naive GEBLNet model without TrNorm layers can learn the Chern number for up to 3 bands but fails for 4 bands and above, mostly outputting zero local quantities except for a few random seeds. Analyzing the statistics of output traces (Figure 4) reveals that variance accumulates across layers, reaching $10^5$ after the final GEAct layer, causing numerical instability and vanishing gradients.

Introducing TrNorm layers mitigates this issue, stabilizing the variance and enabling the model to learn Chern number of systems with more than 4 bands. Figure 3 compares the loss curves between models with and without TrNorm layers, demonstrating the effectiveness of this modification.

Furthermore, training solely on trivial samples limits the model to learning the Chern number up to a global, sample independent, rescaling factor, i.e. $f(W) \approx k\tilde{C}$ for some $k$. To evaluate on general samples, we compute the rescaling factor $R_{\text{scale}} = \text{mean}(\tilde{C})/\text{mean}(f(W))$ using a large set of non-trivial samples and scale the output as $R_{\text{scale}}f(W)$. The training results demonstrated prediction accuracy comparable to that of training on general datasets, as detailed in Table 3 in the Appendix, we obtain an accuracy of approximately $94.1\%$ on four bands, comparable to $95.9\%$ for training on data which includes non-trivial Chern numbers. Meanwhile, Figure 6a shows the model's ability to capture local quantities accurately.

**Larger grids** Due to its local structure, GEBLNet can process samples of arbitrary grid sizes. We evaluate its generalization abilities by testing samples on larger grids using a representative

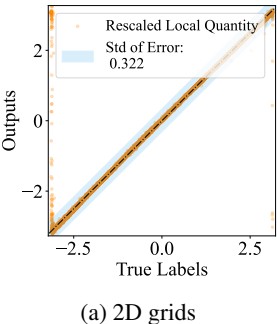
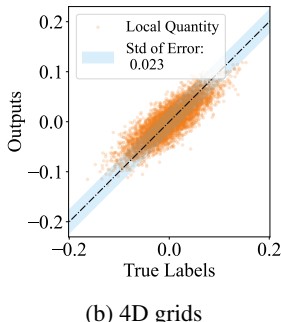

| (a) 2D grids | (b) 4D grids |
|:---:|:---:|

Figure 6: Comparison of rescaled local outputs with local true values. Points closer to the reference line $y = x$ indicate higher accuracy in capturing local quantities.

model trained on smaller grids. The results, shown in Figure 5, indicate that our models show excellent generalization ability to larger grid sizes. The moderate accuracy decrease we observe is approximately linearly with the number of grid sites, likely due to accumulating errors. Training on larger grids slightly improves performance but comes at a considerable computational cost.

**Learning higher-dimensional Chern numbers** We extended the task to learning Chern numbers to $4D$ grids, whose definitions are significantly more complex than the $2D$ case (see Appendix B). Furthermore, instead of a single flux $W_k$, there are $C_4^2 = 6$ Wilson loops per site in $4D$. Since for higher dimensions, the discrete approximation to the Chern number is not necessarily an integer, we cannot use the accuracy for evaluation. Therefore, we use the MAE (global loss $L_g$) instead, see Figure 10 in the Appendix. With an MAE of around $0.25$, our models can predict these higher-dimensional Chern numbers well within rounding errors. Figure 6b demonstrates the predictions of local quantities, showing good agreement with the targets.

## 7   Conclusions and Limitations

In this paper we have introduced a gauge equivariant model that can learn Chern numbers of certain simplistic topological insulators. A limitation of our work is that we only consider Chern numbers and not other topological invariants. We expect that our model can be adapted to a more general setting, and thereby deal with other interesting invariants. In fact, our construction may be viewed as a toy model for more interesting physical systems. Since even learning the Chern number is challenging, this is a stepping stone towards more sophisticated condensed matter systems exhibiting richer topological properties.

## Acknowledgments

We want to thank David Müller and Daniel Schuh for inspiring discussions. The work of J.G. and D.P. is supported by the Wallenberg AI, Autonomous Systems and Software Program (WASP) funded by the Knut and Alice Wallenberg (KAW) Foundation. M.G. acknowledges support from KAW through the Wallenberg Centre for Quantum Technology (WACQT). The computations were enabled by resources provided by the National Academic Infrastructure for Supercomputing in Sweden (NAISS), partially funded by the Swedish Research Council through grant agreement no. 2022-06725.

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

## A  Physical Background

In this section, we give a brief overview of the physics of topological insulators and show how the expression (2) arises as a discretization of the contiuous Chern number.

The band structure of the topological insulators we want to consider here is described by so-called Bloch Hamiltonians $H(k)$ which are maps from the Brillouin zone to the space of $M \times M$ complex Hermitian matrices. The Brillouin zone is the space of momenta $k$ of the electrons which is periodic in each dimension. At each point $k$ in the Brillouin zone, we consider the eigenvectors $v_n(k) \in \mathbb{C}^M$, $n = 1, \ldots, N$, of the Bloch Hamiltonian with negative eigenvalues since these correspond to the bands occupied by electrons in the material.

A nontrivial Chern number means that it is impossible to find smoothly varying eigenvectors over the entire Brillouin zone. In two dimensions, it is defined via

$$C = \frac{1}{2\pi i} \int_{BZ} \text{Tr}\left[\mathcal{F}(k)\right] d^2 k \,, \tag{10}$$

where $\mathcal{F}$ is an $N \times N$-matrix known as the non-abelian Berry curvature defined by

$$\mathcal{F} = \partial_{k_x}\mathcal{A}_y(k) - \partial_{k_y}\mathcal{A}_x(k) + [\mathcal{A}_x(k), \mathcal{A}_y(k)] \,. \tag{11}$$

Here, $\mathcal{A}_\mu(k)$, $\mu = x, y$ is the non-abelian Berry connection, another $N \times N$-matrix whose components are $\mathbb{C}^2$-vectors given by

$$[\mathcal{A}_\mu(k)]_{n,m} = v_n(k)^\top \partial_{k_\mu} v_m(k) \tag{12}$$

in terms of the eigenvectors of the Bloch Hamiltonian with negative eigenvalues.

It can be shown that the Chern number defined by (10) is an integer and there are generalizations to higher dimensional Brillouin zones, on which we performed experiments, see Section 6.3.

The link matrices $U_{i,j}^x, U_{i,j}^y \in \mathbb{C}^{N \times N}$ in terms of which the Wilson loops are constructed according to (1) capture the overlap between the eigenvectors of the Bloch Hamiltonians of neighboring grid points and have components

$$[U_{i,j}^x]_{m,n} = v_m(k_{i,j})^\top v_n(k_{i-1,j}) \tag{13}$$

$$[U_{i,j}^y]_{m,n} = v_m(k_{i,j})^\top v_n(k_{i,j-1}) \,. \tag{14}$$

The links are discrete analogous of the operator $\exp(i\mathcal{A}(k)dk)$. Under gauge transformations, the links transform according to

$$U_{i,j}^x \to \Omega_{i,j}^\dagger U_{i,j}^x \Omega_{i,j} \,, \qquad U_{i,j}^y \to \Omega_{i,j}^\dagger U_{i,j}^y \Omega_{i,j} \,. \tag{15}$$

The Wilson loops correspond to closed $1 \times 1$ loops of the link variables. Their gauge transformation $W_{i,j} \to \Omega_{i,j}^\dagger W_{i,j} \Omega_{i,j}$ follows from the transformation (15) of the links. In higher dimensions, there are several Wilson loops $W_k^\gamma$ per grid point $k$ which are aligned with different directions $\gamma$ in the lattice.

## B  Higher Order Chern Numbers

In 3.1, we defined in (2) for two dimensional Brillouin zones the non-abelian Berry curvature as

$$\mathcal{F} = \partial_{k_x}\mathcal{A}_y(k) - \partial_{k_y}\mathcal{A}_x(k) + [\mathcal{A}_x(k), \mathcal{A}_y(k)] \,.$$

For $2n$ dimensional Brillouin zones, which are topologically equivalent to $\mathbb{R}^{2n}/_{\mathbb{Z}^{2n}}$, there are $P_{2n}^2$ different oriented planes, i.e. for every two directions $k_\mu, k_\nu$, there is a planar flux

$$W_k^{\mu,\nu} = U_k^\mu U_{k+\hat{\mu}}^\nu (U_{k+\hat{\nu}}^\mu)^\dagger (U_k^\nu)^\dagger. \tag{16}$$

It is easy to verify that $W_k^{\mu,\nu} = (W_k^{\nu,\mu})^\dagger$. Similarly, there is a planar curvature

$$\mathcal{F}_{\mu,\nu} = \partial_{k_\mu}\mathcal{A}_\nu(k) - \partial_{k_\nu}\mathcal{A}_\mu(k) + [\mathcal{A}_\mu(k), \mathcal{A}_\nu(k)] \,. \tag{17}$$

Where $\mathcal{A}_\mu$ is similarly defined as in (12). We showed in 3.1 the definition of Chern numbers on a $2D$ Brillouin zone in (10). For a $2n$ dimensional Brillouin zone, a $n_{th}$ order Chern number is defined as

$$C_n = \left(\frac{1}{2\pi i}\right)^n \int_{BZ} \text{Tr}\left[\mathcal{F}(k)^n\right] d^{2n}k. \tag{18}$$

Here, $\mathcal{F}(k)^n$ represents a wedge product of differential forms $\mathcal{F}_{\mu,\nu}(k)\, dk_\mu\, dk_\nu$, which could be written equivalently as

$$\frac{2^n n!}{(2n)!} \sum_{\mu_1,\mu_2,\ldots,\mu_{2n-1},\mu_{2n}} \epsilon_{\mu_1,\mu_2,\ldots,\mu_{2n-1},\mu_{2n}} \prod_{t=1}^{n} \mathcal{F}_{\mu_{2t-1},\mu_{2t}}(k). \tag{19}$$

It could be shown that $C$ is always an integer, $\forall n \geq 1$.

In practice, since the fluxes $W_k^{\mu,\nu}$ is an approximation of $\exp(\mathcal{F}_{\mu,\nu})$, we calculate the discrete version of higher order Chern numbers with the following equation

$$\tilde{C}_n = \frac{n!}{(2n)!(\pi i)^n} \sum_k \sum_{\mu_1,\mu_2,\ldots,\mu_{2n-1},\mu_{2n}} \text{Tr}\epsilon_{\mu_1,\mu_2,\ldots,\mu_{2n-1},\mu_{2n}} \prod_{t=1}^{n} \log W_k^{\mu_{2t-1},\mu_{2t}}. \tag{20}$$

When taking $n = 1$, Equation (20) coincides with (2). Since $\log$ function is analytical, which means could be represented by a power series, and $(\Omega^\dagger W \Omega)^n = \Omega^\dagger W^n \Omega$, we have

$$\tilde{C}(W_k^{\mu_{2t-1},\mu_{2t}}) = \tilde{C}(\Omega^\dagger W_k^{\mu_{2t-1},\mu_{2t}} \Omega), \ \forall \Omega \in U(N)$$

This discretized Chern number is an integer only in the continuum limit, therefore we use the MAE (global loss $L_\text{g}$) instead for evaluation.

## C  Data Generation

### C.1  Uniform Distribution on $U(N)$ with QR Decomposition

In the experiments we generate $U(N)$ with QR Decomposition on a matrix $A \in \mathbb{C}^{N \times N}$, whose entries have i.i.d. $\mathcal{N}(0,1)$ real and imaginary parts. We assume the algorithm to generate $U$ from $A$ is single-valued, i.e. $U = f(A)$ for some function $f : \mathbb{C}^{N \times N} \to U(N)$. We show the "left invariance" of the random variable $U$.

**Proposition C.1.** *$U$ and $gU$ are identically distributed, $\forall g \in G$.*

*Proof.* By definition, $gU = f(gA)$. Then it suffices to show $gA$ and $A$ are identically distributed.

For complex matrices, we consider the two bijections. The first one is $p : A \to \begin{pmatrix} \text{Re}A \\ \text{Im}A \end{pmatrix}$. Then $p(gA) = \begin{pmatrix} \text{Re}g & -\text{Im}g \\ \text{Im}g & \text{Re}g \end{pmatrix} = \hat{g}p(A)$. It is easy to verify $\hat{g}$ is orthogonal. We then flatten the matrix with a vec operator

$$\text{vec}(A) = (A_{11}, A_{12}, \ldots, A_{1N}, \ldots, A_{M1}, \ldots, A_{MN})$$

The following property is well known.

**Proposition C.2** (Vec Operator Identity)**.** *$vec(gA) = (g \otimes I_N)vec(A)$*

Where $\otimes$ is the Kronecker product. Then $\text{vec}(p(gA)) = (\hat{g} \otimes I_N)\text{vec}(p(A))$. However, $\text{vec}(p(A))$ is just $(\text{Re}A_{ij}, \text{Im}A_{ij})$, which follows the distribution $\mathcal{N}(0, I_{2N^2})$, and $\hat{g} \otimes I_N$ is still orthogonal, it follows that

$$\text{vec}(p(gA)) \sim \mathcal{N}(0, (\hat{g} \otimes I_N)I_{2N^2}(\hat{g} \otimes I_N)^T) = \mathcal{N}(0, I_{2N^2})$$

Therefore $\text{vec}(p(gA)) \sim \text{vec}(p(A))$. By bijectivity, $gA \sim A$. $\qquad \square$

## C.2 Diagonal Dataset

There is a equivalance relation among samples $W_k$: $W_k \sim \Omega_k \tilde{W}_x \Omega_k^\dagger, \forall \Omega_k \in U(n), \forall k$. Namely the equivalent classes of fluxes is a subset of $(U(n)/\text{Ad})^{N_\text{site}}$. By the isomorphism

$$U(n)/\text{Ad} \cong U(1)^n/S_n, \tag{21}$$

we could generate plaquettes $W_k$ as diagonal matrices, i.e. $W_k = \text{diag}\{e^{i\theta_k^1}, \dots, e^{i\theta_k^N}\}$.
Notice that each link appears exactly twice in all plaquettes, once in itself, and once inversed. For example, $U_k^x$ appears in itself in $W_k$ and inversed in $W_{k-\hat{y}}$. Then we have:

$$\prod_k \det W_k = \prod_{\mu,k} \det U_k^\mu (\det U_k^\mu)^{-1} = 1$$

Specifically, since $\sum_k \text{Im}(\log(\det W_k)) = \text{Im}(\log(\prod \det W_k)) \mod 2\pi$, the discrete Chern number $\tilde{C}$ is an integer.

**Proposition C.3.** $\tilde{C} = \frac{1}{2\pi} \sum_x F_x = n \in \mathbb{Z}$.

Then the necessary condition for a set of plaquettes to be generated from some links is:

$$\prod_k \prod_\lambda e^{i\theta_k^\lambda} = e^{i \sum_k \sum_\lambda \theta_k^\lambda} = 1, \tag{22}$$

On the other hand, given any $W_k$ that is diagonal per site, suppose it is generated by diagonal links $U_k^\mu = \text{diag}\{e^{i\tau_{k,\mu}^1}, \dots, e^{i\tau_{k,\mu}^N}\}$. Then for each index $\lambda$ we have the following equations:

$$\prod e^{i\tau_{k,x}^\lambda} e^{i\tau_{k+\hat{x},y}^\lambda} e^{-i\tau_{k+\hat{y},x}^\lambda} e^{-i\tau_{k,y}^\lambda} = 1, \forall k \tag{23}$$

This implies a necessary condition for $W_k$ to be generated from diagonal links is that, for any $\lambda$, $\sum \theta_k^\lambda = 0$. We omit the subscript $\lambda$ for now.
Recall that $k$ is the flattened index of $(i,j)$, which could have the possible form $k = N_\text{site} i + j$. If we further flatten the index $(k, \mu)$ as $k$ for $\mu = x$, $k + N_\text{site}$ for $\mu = y$, then the equations become linear:

$$\tau_k + \tau_{(k+N_\text{site}+1) \mod 2N_\text{site}} - \tau_{(k+N_x) \mod N_\text{site}} - \tau_{k+N_\text{site}} = \theta_{\hat{x}}, \forall k \tag{24}$$

Which is just:

$$\begin{pmatrix} 1 & & -1 & & & -1 & 1 & & \\ & \ddots & & \ddots & & & \ddots & \ddots & \\ -1 & & \ddots & & -1 & & & \ddots & \ddots \\ & \ddots & & \ddots & & & & \ddots & 1 \\ & & -1 & & 1 & 1 & & & -1 \end{pmatrix}^T \begin{pmatrix} \tau_0 \\ \tau_1 \\ \vdots \\ \tau_{2N_\text{site}-1} \end{pmatrix} = \begin{pmatrix} \theta_0 \\ \theta_1 \\ \vdots \\ \theta_{N_\text{site}-1} \end{pmatrix}$$

$$\tag{25}$$

The coefficient matrix has rank $N_\text{site} - 1$, and it is solvable iff. $\sum_k \theta_k = 0$, and that is exactly what the necessary condition specifies. Therefore, the fluxes $W_k$ can be generated from diagonal $U_k^\mu$ if and only if

$$\forall \lambda, \prod_k e^{i\theta_k^\gamma} = 1. \tag{26}$$

This determines a submanifold $M'$ in $M = \{m \in U(1)^{N \times N_\text{site}} : m \text{ satisfies } (22)\}$ with codimension $N - 1$. With the natural metric on $U(N)^{N_\text{site}} \supset M$, defined as $d(g,h) = \|\psi_k^\lambda\|_2$, where $\psi_k^\lambda$ are phase angles of eigenvalues of $gh^{-1}$, $M'$ is a $\pi\sqrt{\frac{N}{N_\text{site}}}$-net of $M$. For each channel $\lambda$, suppose $\sum_k \theta_k^\lambda = \phi_\lambda$, $\phi_\lambda \in [-\pi, \pi)$. Let the new $\theta$ be $\tilde{\theta}_k^\gamma = \theta_k^\lambda + -\phi_k/N_\text{site}$. Then

$$d(W, \tilde{W}) \le \sqrt{\sum_{k,\lambda} \left(\frac{1}{N_\text{site}}\right)^2 \phi_k^2} \le \pi\sqrt{\frac{N}{N_\text{site}}}.$$

As the number of sites gets larger (the grid gets more refined), the net gets denser. We can further extend the sufficient condition by considering the permutations, since the permutation matrices are also unitary and their actions on fluxes are adjoint.

We now propose the diagonal data generation scheme:

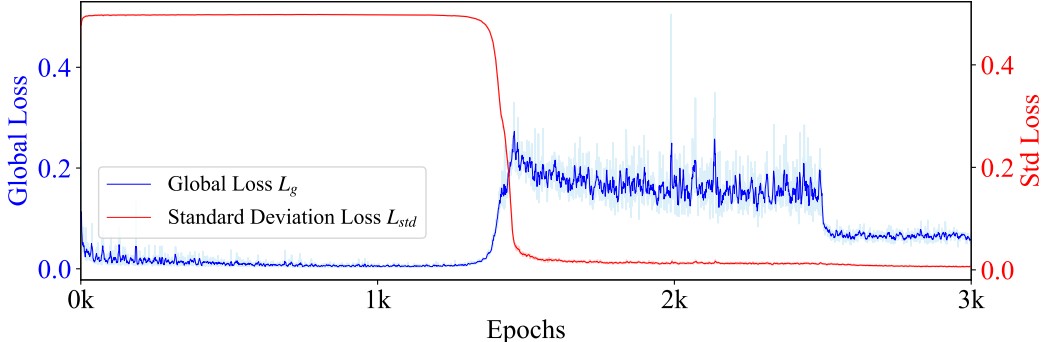

Figure 7: Global Loss and Standard Deviation Loss curve of the baseline model, trained on a diagonal, trivial dataset, to learn the Chern number on a $5^2$ grid, with $4$ filled bands.

Table 3: Accuracy of the same run in Figure 7, evaluated on non-diagonal, non-trivial data on a $5 \times 5$ grid, with $4$ filled bands.

| Seeds | No.1 | No.2 | No.3 | No.4 |
|---|---|---|---|---|
| Accuracy | $92.7\%$ | $94.3\%$ | $95.4\%$ | $93.8\%$ |

1. Generate label $F_k \in [-\pi, \pi)$, such that $\sum F_k = 2\pi n$.
2. If only zero samples: check if $\sum F_k = 0$.
3. For every $k$ but the last one, generate $(\phi_k)_x$ such that $\sum_\lambda \phi_k^\lambda = F_k$.
4. For every $k$ but the last one, let $W_k$ be $\mathrm{diag}\{e^{i\theta_k^1}, \ldots, e^{i\theta_k^N}\}$.
5. Let the last $W_{\hat{k}}$ be $\prod_{k \neq \hat{k}} W_k^{-1}$.

The last product will not cause confusion since diagonal matrix multiplication is commutative.

It could also go the other way around: generate the fluxes first, then find a solution to (25) to get the links. This way, we could operate directly on the distribution of eigenvalues, thus customizing the data generation process. Furthermore, the diagonal dataset reduces the computation cost significantly for training.

For validation, we show in Figure 7 the loss curves and in Table 3 accuracies of evaluation on nontrivial, general (non-diagonal) datasets, of a training run on a diagonal, trivial dataset.

# D Proof of the Universal Approximation Theorem

Here we give the complete proof of Theorem 5.2, or, rather its stronger form in Theorem D.3. We begin with some properties of the class functions defined in 5.1.

**Theorem D.1.** *The space of symmetric polynomials $\{s_k\}$ over eigenvalues $\{\lambda_k\}$ forms an orthonormal basis of $L^2_{class}(G)$.*

*Proof.* Follows directly from the Peter-Weyl theorem and the expansion of class functions in terms of irreducible characters. □

Furthermore, consider the set of polynomials over eigenvalues of $g$: $\{p_k(\lambda_1, \ldots, \lambda_n) = \sum_i \lambda_i^k\}$. In our setting these polynomials can be identified with the set of traces of group elements of the form $\mathrm{Tr}g^k$. Using Newton's identities for symmetric polynomials:

$$ke^k = \sum_{j=1}^{k}(-1)^{j-1}e_{k-j}p_j, \quad e_k = \sum_{\sum_n k_n = k}\prod \lambda_i^{k_i} \tag{27}$$

one may deduce the following

**Corollary D.2.** *The space $\bigcup_M \{f(Trg, \ldots, Trg^M)\} \cap L^2(G)$ is dense in $L^2_{class}(G)$.*

Now, we consider the network architecture GEBLNet. Given the flux tensor $W_k$, we stack the identity and its Herimitian conjugate to a second channel as

$$W_k'^{\gamma} = (W_{k,0}, W_{k,1}, W_{k,-1}) := (I, W_k, W_k^{-1}).$$

Afterwards we put it through several blocks, each containing three layers: GEBL, GEAct and TrNorm. In this section, we ignore TrNorm Layers, since they are introduced to boost training results. We call each block a **"packed layer"**.

After several packed layers we calculate the trace per-channel and add a linear layer (the "Dense layer") in the end. The Dense layer acts on the real and imaginary parts separately. Then we take the sum over the site index (to calculate the topological invariant).

So the outputs have the following form:

$$W_k \longmapsto w \cdot \hat{\sigma} \circ \mathrm{GEBL}_n \circ \cdots \circ \hat{\sigma} \circ \mathrm{GEBL}_1(W_k)) + b.$$

Where $\hat{\sigma}(W_k^{\gamma}) = \sigma(\mathrm{Tr}W_k^{\gamma})W_k^{\gamma}$. We denote the set of these functions by $\mathcal{BLN}_{\sigma}(G)$, where the subscript $\sigma$ indicates the choice of activation function. We further denote by $\mathcal{BLN}_{\sigma}^k(G)$ the subset of $\mathcal{BLN}_{\sigma}(G)$ with $k$ packed layers.

Since we attempted to learn local quantities $F(W_k)$, we omit the subscript $k$. Furthermore, we treat the flux $W$ as an abstract element in the Lie group $G$, denoted as $g$. In this case, where the input channel size is one, we propose the main result:

**Theorem D.3** (Universal Approximation Theorem). *For any activation function $\sigma = \tilde{\sigma} \circ Re$, where $\tilde{\sigma}$ is bounded and non-decreasing, $\mathcal{BLN}_{\sigma}(G)$ is dense in $L^2_{class}(G)$.*

The proof of this will require the following lemma.

**Lemma D.4.** *$\mathcal{BLN}_{\sigma}^k(G)$ is dense in $\{f(p_1, \cdots, p_{2^k}) : \|f\|_{\infty} < \infty\} \subset L^{\infty}(G)$, where $p_i = Trg^i$.*

*Proof.* We prove this lemma by induction. For $k = 1$, the output has the following form

$$g \longmapsto \left(\sum_{i=0}^{2} \alpha_i^t g^t\right)_i \quad \longmapsto \quad \omega_i \mathrm{Tr}\sigma\left(\sum_{i=0}^{2} \alpha_i^t g^t\right)_i + b.$$

Note that $\mathrm{Tr}\hat{\sigma}(\sum_{i=0}^{2} \alpha_i^t g^t) = \sigma(\mathrm{Re}\alpha_i^t p_t)\alpha_i^t p_t$. For any channel index $i$, when taking only the real part (in other words, forcing $w_{i,\mathrm{Im}}$ in the dense layer to be zero), the output is simply

$$\sigma\left(\sum_t \mathrm{Re}\alpha_i^t \mathrm{Re}p_t - \mathrm{Im}\alpha_i^t \mathrm{Im}p_t\right)\left(\sum_t \mathrm{Re}\alpha_i^t \mathrm{Re}p_t - \mathrm{Im}\alpha_i^t \mathrm{Im}p_t\right) = \hat{\sigma}\left(\sum_i \mathrm{Re}\alpha_i^t \mathrm{Re}p_t - \mathrm{Im}\alpha_i^t \mathrm{Im}p_t\right) \tag{28}$$

Therefore, it is essentially a one-hidden-layer fully connected network on $\{(p_1, p_2)\} \simeq \mathbb{R}^4$. Thus the set is dense.

Assume this is the case for $n$, and we would like to prove the lemma for $n + 1$. We denote $2^n = N$. Then the layer input has the following form:

$$\tilde{\sigma} \left( \sum_{t=0}^{N} a_i^t(p_0, \cdots, p_{N/2}) p_t \right) \left( \sum_{t=0}^{N} a_i^t(p_0, \cdots, p_{N/2}) g^t \right),$$

Now the new "$a_i^t$"(denoted as $b_i^t$) takes the following form:

$$b_i^t = \sum_{p+q=t} \sum_{j,k} \alpha_{ijk} \tilde{\sigma}(a_j^t p_t) \tilde{\sigma}(a_k^t p_t) a_j^p a_k^q.$$

Consider the bijection $F : \mathbb{C}^{N+1} \to \mathrm{P}_N(\mathbb{C})$, given by $F(\vec{a}) = \sum_t a_t z^t$. Using this we define

$$\vec{a} * \vec{b} = F^{-1}(F(\vec{a})F(\vec{b})).$$

Then

$$\vec{b_i} = \alpha_{ijk} \tilde{\sigma}(\vec{a_j} \cdot \vec{p}) \tilde{\sigma}(\vec{a_k} \cdot \vec{p}) \vec{a_j} \vec{a_k} = \alpha_{ijk} H(p, \vec{a_j}, \vec{a_k}).$$

This forms a linear space $\mathcal{B}^{n+1} \subset (L^\infty(K_{n+1}))^{2N+1}$. For simplicity we henceforth omit the subscript on $K_{n+1}$.

We assume $(a_i^t)_{t=0}^N$ could approximate any constant function of $p_1, \cdots, p_{N/2}$. This is trivially true when $n = 1$, since it is a function on a constant and takes arbitrary constant values.

Denoting $e_0 = F^{-1}(1/d)$, where $d = \dim G$, we have $e_0 \cdot p = 1, \forall p \in K$. Since $K$ is compact, there exists an open set $U$ s.t. $e_0 \in \partial U$, and $b \cdot p \in (1, +\infty), \forall b \in U, p \in K$.

On the other hand, it is easy to see that $\{b * b : b = (1, z, \cdots, z^N)\}$ is linearly independent as a subset. This way we could choose $2N + 1$ elements $\{b^{z_t}\}_{t=0}^{2N}$ from its intersection with $U$, such that $\mathrm{span}\{b^{z_t} * b^{z_t}\} = \mathbb{C}^{2N+1}$.

Now given a constant vector $\vec{b} = (b_0, \cdots, b_{2N})$, there exits $\{\alpha_t\}$ such that $\vec{b} = \alpha_t b^{z_t} * b^{z_t}$. We want to show that $\vec{b}$ can be approximated by any precision $\epsilon$.

Without loss of generality, assume $\sup \tilde{\sigma} = 1$ and $\inf \tilde{\sigma} = 0$. Then, for all $\epsilon$, there exists $M_0 > 0$ such that for all $x > M_0/2$, $\tilde{\sigma}(x) \in (\sqrt{1-\epsilon}, 1)$. This gives

$$\left| \frac{d^2}{M^2} H(p, Me_0, Me_0) - 1 \right| = |1 - \tilde{\sigma}(M)^2| < \epsilon, \quad \forall M > M_0.$$

By induction, there exists $a_t$ such that $\|a_t - b^{z_t}\|_\infty < \min\{\epsilon, M/2\}$. Consider

$$\vec{b'} = \alpha_t \frac{1}{M^2} H(p, a_t, a_t) = \alpha_t \tilde{\sigma}(M\alpha_t \cdot p)^2 a_t * a_t.$$

Then

$$
\begin{aligned}
|\vec{b'} - \vec{b}| &= |\alpha_t(\tilde{\sigma}(M\alpha_t \cdot p)^2 - 1)b^{z_t} * b^{z_t} + \tilde{\sigma}(M\alpha_t \cdot p)(b^{z_t} * b^{z_t} - \alpha_t * \alpha_t)| \\
&\leq \alpha_t \epsilon |b^{z_t} * b^{z_t}| + 2\epsilon|b^{z_t}| + \epsilon^2 \\
&\leq C(\vec{b}, N)\epsilon.
\end{aligned}
\tag{29}
$$

When the coefficient functions approximate constants, the last layer is essentially a one-hidden-layer fully connected network over $p_1, \cdots, p_{2N}$. Similar to the $N = 1$ case, as the width grows larger the network can approximate any function $f(p_1, \cdots, p_{2N})$. the concludes the proof of the lemma. $\square$

We may now complete the proof of Theorem D.3.

*Proof.* (Proof of Theorem D.3)
Recall that by Theorem D.1 the space of class functions $L_{class}^2(G)$ is spanned by symmetric polynomials in the eigenvalues of group elements. Since these symmetric polynomials can be expressed

in terms of traces $\mathrm{Tr}(g), \mathrm{Tr}(g^n), \ldots, \mathrm{Tr}(g^M)$ it follows that any class function can be written as a function of these traces. Now, since $G$ is compact, we have $L^2(G) \supset L^\infty(G)$ and $\|f\|_2 \geq C\|f\|_\infty$. Therefore, for all $f \in L^2_{\mathrm{class}}(G)$, and for any $\epsilon > 0$, there exists

$$f_n = f_n(p_1, \cdots, p_n) \in L^\infty(K)$$

such that $\|f - f_n\|_2 < 1/2\epsilon$. By Lemma D.4 the function class $\mathcal{BLN}_{\hat{\sigma}}^k(G)$, consisting of neural networks with $k$ gauge equivariant bilinear layers, can approximate any function $f(p_1, \ldots, p_k)$ arbitrarily well, provided $k$ is large enough. We deduce that there exists $g \in \mathcal{BLN}_{\hat{\sigma}}^n(G) \subset L^\infty(G)$ such that $\|g - f_n\|_\infty < 1/2C\epsilon$. Therefore

$$\|g - f\|_2 < (C \cdot 1/2C + 1/2)\epsilon = \epsilon.$$

This concludes the proof of the main theorem. $\qquad\square$

Restricting our scope to unitary groups, we could further generalize to the case where the input data has more than one channels. In other words, the function $f(g_1, \ldots, g_K)$ we attempt to learn has the following property,

$$f(gh_1g^{-1}, \ldots, gh_Kg^{-1}) = f(h_1, \ldots, h_K), \forall g, h_i \in G. \tag{30}$$

For inputs with $n$ channels, we denote the set of functions that could be represented by GEBLNet as $\mathcal{BLN}_{\hat{\sigma}}(G^n)$.

**Theorem D.5.** *For any activation function $\hat{\sigma}$ that is bounded and non-decreasing, $\mathcal{BLN}_{\hat{\sigma}}(G^n)$ could approximate any function with the property specified in* (30) *in the $L^2$ norm.*

The proof requires the following theorem in [40].

**Theorem D.6** (Procesi, 1976). *Let $F$ be a field of character $0$, and let*

$$F\big[A_k = (x_{ij})^{(k)}_{N \times N} \mid 1 \leq k \leq K\big]$$

*be the polynomial ring in $KN^2$ variables, corresponding to scalar-valued polynomials on $F^{N \times N \times K}$. Then the subalgebra*

$$\big\{p \in F[A_k] \mid p(A_1, \ldots, A_K) = p(gA_1g^{-1}, \ldots, gA_Kg^{-1}), \ \forall g \in GL_N(F)\big\} \tag{31}$$

*is generated by $\big\{\mathrm{Tr} \prod_{k=1}^K A_{i_k}^{n_k}\big\}$.*

*Proof of Theorem D.5.* It is well-known that the unitary group is not contained in the root set of any nonzero polynomial on the set of $n$ by $n$ matrices $M_n(\mathbb{C})$. Therefore, for an arbitarily chosen sequence of $(A_1, \ldots, A_k)$ in $G$, if for any $g \in G$ we have $p(A_1, \ldots, A_k) = p(gA_1g^{-1}, \ldots, gA_kg^{-1}), \forall g \in G$, then this also holds for any $g \in M_n(\mathbb{C})$. Analogously we could show that the equation in Eq. (31) is equivalent to Eq. (30) for polynomials. Namely, functions satisfying Eq. (30) are spanned by $\big\{\mathrm{Tr} \prod_{k=1}^K A_{i_k}^{n_k}\big\}$, and the proof, analogous to the one-dimensional case, follows. $\qquad\square$

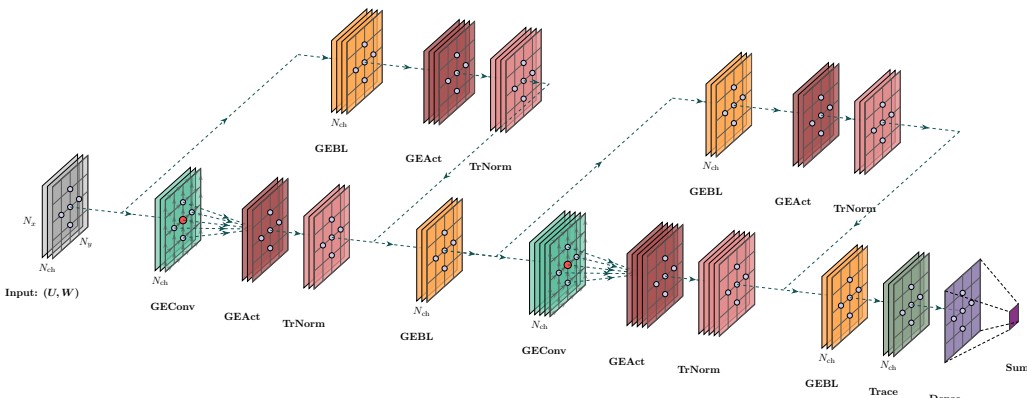

Figure 8: Architecture of GEConvNet. In this figure, the rectangles represent the spatial grid, the arrows on the grid represent links, and the number of layers ($N_{ch}$) represents the number of channels ($\gamma$). Each circle represents a site on the grid, and quantities on different sites do not interact with each other, except for the GEConv Layers, and the last summation on grids.

Table 4: Configuration of GEBL layers for the representative model. See its architecture in Figure 2. In addition, every GEBL layer is followed by a GEAct layer and a TrNorm layer that maintains channel size.

| Layer | Input Channel | Output Channel |
|-------|:-------------:|:--------------:|
| GEBL 1 | 1 | 32 |
| GEBL 2 | 32 | 16 |
| GEBL 3 | 16 | 8 |

# E   Further Details regarding the Network

**Architecture of GEConvNet**   Figure 9 is a demonstration of the architecture of GEConvNet. It has GEConv layers implemented inside, therefore it takes the links $U_k^\mu$ as inputs, beside the fluxes $W_k$, if the kernel size is set as positive.

**Configuration of the representative model**   Table 4 lists the hyperparameters for GEBL layers in the representative model. Since the GEBL layers are consecutive with channel size-maintaining layers, GEAct and TrNorm layers in between, the former layer's output channel equals the latter's input channel.

**Complexity-Accuracy Comparison**   Figure 9 is a comparison of model complexity and accuracy across different models. All the models are trained on a $5 \times 5$ grid with $4$ filled bands. Due to the instability of TrMLP, there are only 3 successfull runs with this model.

**General Compute Resource Requirements**   All experiments were conducted on a computing cluster equipped with NVIDIA T4 GPUs. For the baseline task of training GEBLNet to predict Chern numbers on a $5 \times 5$ grid, training typically required approximately 15 to 20 hours with 1 T4 GPU and 32 CPU cores. The same hardware resources are sufficient to train models for tasks on $4D$ systems or on larger grids; however, the computational time generally increases with the task complexity. While it is possible to train on personal computers, we do not provide guarantees for the feasibility of such setups.

**Global Loss Curve for Training on Higher Order Chern Numbers**   Figure 10 shows the global loss curve for training on second order Chern numbers. Since we adopt the $L_1$-norm $\|\cdot\|_1$ here, $L_g$ essentially measures the mean absolute error of global outputs.

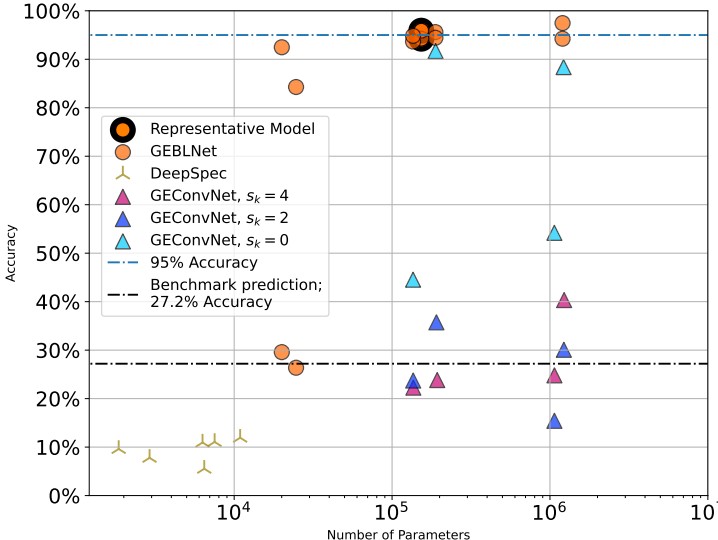

Figure 9: Comparison of model complexity and accuracy across different architectures. Complexity is measured by the number of learnable parameters, and $s_k$ denotes the kernel size for GEConv layers. Each marker represents a training run with variations in models, learnable parameters, and random seeds.

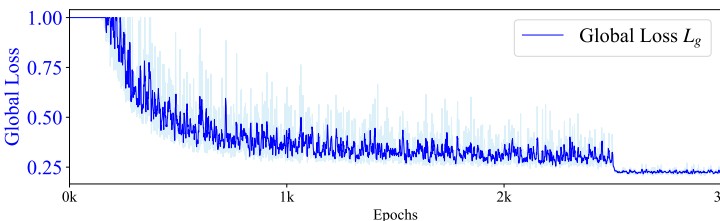

Figure 10: Global validation loss curve of the representative model, trained on a $3^4$ grid, with 3 filled bands to learn the second order Chern number $\widetilde{C}_2$.

# F   Learning Chern numbers using ResNets

Figure 11 shows an ablation over a number of different architectures considered for the task of predicting the determinant of $N \times N$ real matrices.

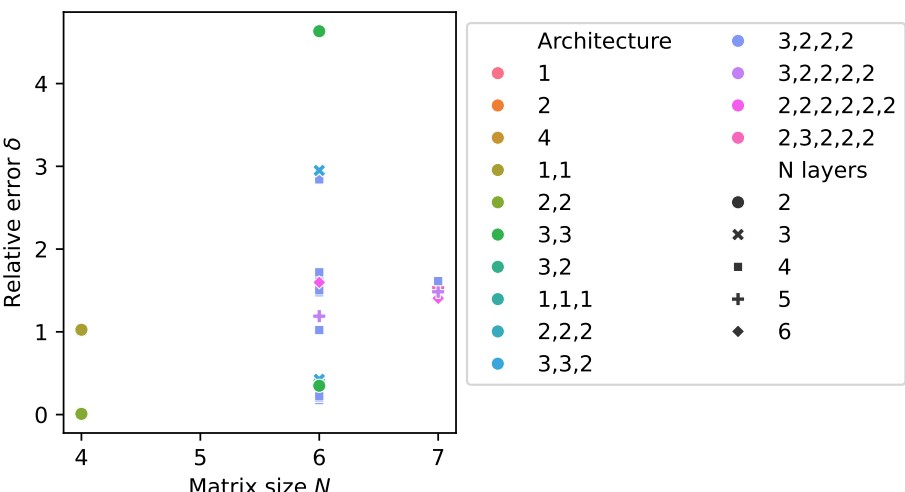

Figure 11: Architecture ablation over different layer orders (see section 3.2) and depths. Best relative error $\delta$ for each architecture for different matrix sizes $N \times N$.

