# OpenReview forum: "Learning Chern Numbers of Multiband Topological Insulators with Gauge Equivariant Neural Networks"
_NeurIPS.cc/2025/Conference — NeurIPS 2025 poster_

### Official Review · Reviewer_cwof · 2025-06-10

**Clarity:** 3
**Significance:** 4
**Originality:** 3
**Rating:** 5
**Confidence:** 4

**Summary:**

The paper introduces a gauge equivariant neural network that allows to predict Chern numbers associated with multiband topological insulators. The model respects local gauge symmetry through the novel gauge equivariant normalization layer as well as some of the established equivariant layers, contributing to the drastic reduction in the size of training dataset. The model is evaluated from multiple aspects and is shown to produce promising results.

**Questions:**

- How is the model for 4D grid implemented?

**Ethical Concerns:**

["NO or VERY MINOR ethics concerns only"]

**Final Justification:**

Topological insulator is actively studied in mathematical physics community. Chern number (or character) is a common mathematical concept, but in general highly abstract notion and not computable at all. I support this paper based on the following two reasons: The authors cleverly use an alternative expression of Chern characters in the context of topological insulator, and the idea to overcome the difficulty in training their proposed models is also clever and mathematically-grounded. The major drawback is, as the authors also agreed (partially), its potential associated computational cost and scalability. However, even when taking the drawback into account, the significance of contribution of the paper outweighs the drawback, and the paper could plays a leading and influential role in the community.

**Limitations:**

- Scalability. The input size is N^2 * Grid size.

**Paper Formatting Concerns:**

Nothing particular.

**Quality:**

3

**Strengths And Weaknesses:**

Strength
- The motivation of incorporation equivariant model is clear
- Benchmarking standard methods, whose results clearly point out downsides or weakness of the relevant methods.
- Solid proof of the universality of the model
- Exhaustive experiments. Some of the ideas to overcome difficulties are based on the authors' deep understanding on the topic, which makes the paper strong. For example, using the fact that each orbit has a diagonal matrix for the dataset generation is clever.

Weakness
- The importance of the Chern number in the context of topological insulators is not very clearly stated.

---

> ### Author Rebuttal · Authors · 2025-07-30
>
> We would like to thank the reviewer for providing such a thorough and positive evaluation of our submission. In the following rebuttal, we will try to answer your questions and clear up potential misunderstandings.
>
> ### Weakness
>
> > • The importance of the Chern number in the context of topological insulators is not very clearly stated.
> >
>
> ### Question
>
> We agree that the physics background was very brief in the manuscript. This was intentional, in order to focus on the mathematical background and the machine learning aspects of the problem. In the following, we will provide some further background about the importance of the Chern number.
>
> Topological insulators are materials that are insulators in the bulk but have conducting states on the surface (edge in the case of a 2D insulator). Topological insulators characterized by a Chern number are the prime example of such materials. The fact that a material with finite Chern number cannot be continuously deformed into a material with Chern number 0 without closing the energy gap (which make the materials insulating) implies that there has to be a zero energy (conducting) state at the boundary. The quantized conductance of the quantum Hall effect is one celebrated physical system which is related to such a conducting boundary mode.
>
> We will add further details to the revised version of the manuscript.
>
> > • How is the model for 4D grid implemented?
> >
>
> The 4D model is implemented analogously to the 2D case. In 2D, the input consists of a single Wilson loop matrix at each lattice site. In 4D, on the other hand, there are $\binom{4}{2} = 6$ independent Wilson loops, each defined on a different 2D subplane. These six Wilson loop matrices serve as the input channels of the model, with gauge transformations acting simultaneously on each of them. Structurally, the 4D model therefore follows the same architecture as the 2D model, but with six input channels instead of one.
>
> ### Limitation
>
> > • Scalability. The input size is N^2 * Grid size.
> >
>
> You are certainly correct in pointing out that scaling of the input size is a potential problem. However, this is precisely a key strength of our gauge equivariant model: since it considers *local* symmetries the scaling problem in grid size is not an issue.

---

> ### Comment · Reviewer_cwof · 2025-08-08
> **Rebuttal acknowledgement.**
>
> I thank the authors for the rebuttal. My concerns were addressed satisfactorily. I will support this paper, because the idea of the paper is very intriguing and inspiring, and believe that the paper will provide a new insight to the material discovery research. The following is a bit of elaboration about the rationale behind my rating.
>
> ---
> Chern character (or number) is a common mathematical concept which has been extensively studied in algebraic topology, but in general this notion is highly abstract and not computable at all. Furthermore, while it is well-known that Chern number plays a crucial role to characterize topological insulators, majority of the relevant existent research is still highly theoretical.
>
> I support this paper based on the following two reasons: The authors cleverly use an alternative expression of Chern characters in an empirical context of topological insulators, which is highly untrivial. Moreover, the idea to overcome difficulties in training their proposed models is also clever and mathematically well-grounded. One drawback could be, as the authors (partially) agreed, its potential high computational cost and relatively small experiments. However, even when taking those drawbacks into account, the significance of the paper's contribution largely outweighs the drawbacks, because the main scope of this paper is to illustrate the usage of an untrivial alternative representation of Chern number and showcase its effectiveness in empirical settings. Bunch of ideas in the paper is inspiring and the paper is worth to be presented in a prestigious venue like NeurIPS to further facilitate future research in the material discovery community.

---

### Official Review · Reviewer_6HNj · 2025-07-03

**Clarity:** 4
**Significance:** 1
**Originality:** 2
**Rating:** 3
**Confidence:** 3

**Summary:**

The authors are able to learn higher-band and higher-dimensional Chern numbers by adding a normalization layer to Lattice Gauge Equivariant CNNs (LGE-CNNs). They prove a universal approximation theorem for the resulting architectures.

**Questions:**

Are there real / existing datasets where applying your new architecture would yield scientifically interesting or instructive results? I will raise my score if you perform such an experiment. Otherwise, I may decrease the score.

**Ethical Concerns:**

["NO or VERY MINOR ethics concerns only"]

**Final Justification:**

Am still not convinced that the current work has sufficient (a) methodological novelty (a normalization layer that is, in the reviewer's opinion, straightforward) , (b) technical novelty, or (c) experimental novelty to merit publication at this time. The authors are encouraged to apply the architecture to more difficult problems to better motivate the methodological usefulness.

**Limitations:**

yes

**Quality:**

3

**Strengths And Weaknesses:**

### Strengths

- The authors are able to fix training stability issues in LGE-CNNs by introducing a new normalization layer.
- Using the introduced architectures, the authors are able to learn Chern numbers with more than 3 bands for the first time.
- Similarly, higher-dimensional Chern numbers are learned.
- The paper is clear, easy to understand. The plots are readable, and the text synthesizes the technical results and experiments quite well.

### Weaknesses

- Relatively little methodological and technical novelty.
    - The main technical contribution is the introduction of a normalization layer. The authors nicely motivate that this is useful to stabilize training. However, because it is a straightforward / incremental patch, it probably shouldn’t merit being listed as a contribution.
    - The main methodological contributions are applications in synthetic datasets. The scientific or technical significance of these constructed tasks is not motivated in the main body.
- The universal approximation theorem is stated as a main theoretical result. However, it uses standard analysis and comparable results already exist for similar equivariant architectures.
- The evaluations are largely toy. These motivate the need for gauge equivariant architectures to learn Chern numbers, but e.g. do not perform new science or have a clear path to doing so.

---

> ### Author Rebuttal · Authors · 2025-07-30
>
> We would like to thank the reviewer for providing such a thorough and detailed evaluation of our submission. In the following rebuttal, we will try to answer your questions and clear up potential misunderstandings.
>
> ### Weaknesses and Questions
>
> > • The main technical contribution is the introduction of a normalization layer. The authors nicely motivate that this is useful to stabilize training. However, because it is a straightforward / incremental patch, it probably shouldn’t merit being listed as a contribution.
> >
>
> We respectfully disagree with the assessment that the normalization layer is merely an incremental patch and not a significant contribution. While normalization is a common technique, numerical instability was a well-known but unsolved issue in the context of gauge equivariant networks, and to the best of our knowledge, no prior work has systematically addressed it.
>
> In particular, our work proposes a tailored normalization mechanism that specifically handles complex-valued representations in this setting. This is not a trivial adaptation of existing techniques:
>
> Instead of standard forms like $\frac{x - \mu}{\sigma}$, which proved ineffective in our setup, we instead divide by the mean. Furthermore, we normalize by the norm of the traces rather than their raw values, respecting the complex valued nature and the gauge symmetry. Other gauge equivariant normalization layers using e.g. the determinant are conceivable but we decided for the current version based on a statistical analysis of the gauge invariants of the preactivations.
>
> We therefore believe this component is both methodological novel, and practically valuable for stabilizing training in this class of networks.
>
> > • The universal approximation theorem is stated as a main theoretical result. However, it uses standard analysis and comparable results already exist for similar equivariant architectures.
> >
>
> Universal approximation has indeed been proven for various globally group equivariant architectures. However, as far as we know, prior to our work no such results existed for *gauge* equivariant architectures (i.e. where the symmetry is *local)*.
>
> > • The main methodological contributions are applications in synthetic datasets. The scientific or technical significance of these constructed tasks is not motivated in the main body.
> >
>
> > • The evaluations are largely toy. These motivate the need for gauge equivariant architectures to learn Chern numbers, but e.g. do not perform new science or have a clear path to doing so.
> >
>
> > Are there real / existing datasets where applying your new architecture would yield scientifically interesting or instructive results? I will raise my score if you perform such an experiment. Otherwise, I may decrease the score.
> >
>
> We appreciate the need for additional motivation for the chosen dataset. As is standard in the literature in this domain (see e.g. [13], [34], [35] as well as [I] and [II]), we train on synthetically generated data. These datasets are based on Hamiltonians which model physical topological materials. However, to date, there is no large-scale dataset of band structures obtained by measuring angular-resolved photoemission spectra which would allow one to train a machine-learning model in this way.
>
> Considering higher-band topological insulators, or generally more complex gauge groups, is an important problem as it opens up for the application of machine learning to related domains such as emergent gauge fields in strongly correlated / topologically ordered phases or artificial non‑Abelian gauge fields in cold atoms and photonics as well as condensed matter physics. Before this work, no machine learning system existed which could predict Chern numbers of higher-band topological materials. Our model significantly extends the SOTA by being able to predict Chern numbers of materials with as much as 7 filled bands.
>
> In particular, we train on uniformly distributed link variables (Appendix C.1) and on link variables whose distribution was adjusted for a specific distribution of Chern numbers (Appendix C.2). These allow us to test our model on input configurations spanning a wide range of Chern numbers with many filled bands, going beyond the limited Hamiltonian models common in the literature. To verify that our model is indeed able to learn also the simpler data from those Hamiltonians, we trained our model on the Bloch Hamiltonian and obtained an accuracy of 95.5%.
>
> References:
> Arabic references correspond to our manuscript.
>
> [I] Che, Y., Gneiting, C., Liu, T., & Nori, F. (2020). *Topological quantum phase transitions retrieved through unsupervised machine learning*. Phys. Rev. B, **102**(13), 134213.
>
> [II] Scheurer, M. S., & Slager, R.-J. (2020). *Unsupervised Machine Learning and Band Topology*. Phys. Rev. Lett., **124**(22), 226401.

---

> ### Comment · Reviewer_6HNj · 2025-08-03
>
> I thank the authors for their detailed response. I remain unconvinced on the empirical and/or algorithmic importance of the work or the novelty or usefulness of the theoretical contribution, and thus will maintain my score. In potential future work, I encourage the authors to show their normalization layer helps make progress on non-toy scientific problems.

---

> > ### Author Response · Authors · 2025-08-06
> >
> > Thank you very much for your feedback on our rebuttal.
> >
> > If you want to elaborate on the reasons for your evaluation, we would be very happy to provide further clarifications.
> >
> > Thank you also for the suggestion to try our normalization layer on different datasets. However, we would like to point out that the normalization layer fixes the unstable training dynamics which appears at moderate depth due to the bilinear layers in the gauge equivariant networks. This instability is therefore independent of the dataset used.

---

> > > ### Comment · Reviewer_6HNj · 2025-08-07
> > >
> > > Thanks for your response.
> > >
> > > > Thank you also for the suggestion to try our normalization layer on different datasets. However, we would like to point out that the normalization layer fixes the unstable training dynamics which appears at moderate depth due to the bilinear layers in the gauge equivariant networks. This instability is therefore independent of the dataset used.
> > >
> > > Presumably, decreasing training instabilities will allow you to train larger models on datasets that are both more difficult and more relevant than the previous state-of-the-art. It does not seem like you properly show this in the current work.

---

> > > > ### Comment · Area_Chair_hHmv · 2025-08-08
> > > >
> > > > I appreciate the detailed rebuttal from the authors and the reviewer’s engagement. After reviewing the exchange, I find that the authors have made a credible case for the technical relevance and novelty of their normalization layer, particularly within the context of gauge equivariant networks. While the reviewer is correct that empirical validation on more applied or real-world datasets would strengthen the paper, the authors reasonably point out that such datasets are not readily available in this subfield and that implementing additional experiments is infeasible within the rebuttal period.
> > > >
> > > > I view the normalization method as a meaningful contribution given the current state of the field, and the synthetic tasks used for evaluation follow established precedent in this literature. While the work could be strengthened by broader empirical validation, I do not believe the absence of such experiments invalidates the paper’s contributions.
> > > >
> > > > I encourage the reviewer to reconsider the weight placed on these additional experiments when forming their final recommendation.

---

### Official Review · Reviewer_FXTi · 2025-07-03

**Clarity:** 3
**Significance:** 3
**Originality:** 3
**Rating:** 5
**Confidence:** 3

**Summary:**

This work presents a new gauge equivariant architecture aimed at learning Chern numbers of a simple model of topological insulators, which are of interest in the condensed matter community. The paper begins by describing the model which (at a very high-level) consists of a 2-dimensional lattice, where each node carries a unitary matrix $W_{i,j} \in U(N)$, where $N$ corresponds to the number of bands. Critically, two instances of this lattice are equivalent up to conjugation by an element $\Omega_{i,j}$ of $U(N)$ which varies from node to node. The task involves predicting a certain formula using these $W_{i,j}$. The paper explains why this problem is hard for more familiar neural networks by exploring network performance on the task of predicting matrix determinants (which is related to the problem above). This motivates the introduction of a family of architectures GEBLNet and GEConvNet that include a novel normalization layer. The paper performs a range of experiments on synthetic data and shows that GEBLNet can successfully learn to predict Chern numbers for different numbers of $N$ and can even by trained to learn on $4$-dimensional lattices.

**Questions:**

- Line 154: The term ‘flux data’ appears here without a definition. It is later referenced several times. What does this mean?
- Reviewer may have missed it, but what is the point of introducing GEConvNet it is not perform particularly well in any of the instances?

**Ethical Concerns:**

["NO or VERY MINOR ethics concerns only"]

**Final Justification:**

The reviewer feels that this is a valuable contribution to the machine learning community. The paper tackles an interesting and scientifically important problem that has so far only been explored to a limited extent in machine learning. This reviewer agrees with the others that the paper probably could have done more to make general readers of the work aware of the scientific and data challenges associated with the problem.

**Limitations:**

Yes.

**Paper Formatting Concerns:**

I have no concerns.

**Quality:**

4

**Strengths And Weaknesses:**

Strengths:
- **The paper is clearly written and should be comprehensible to a significant portion of the NeurIPS community:** The paper does an excellent job making the material intelligible to non-physics readers. Understanding the problem really only requires understanding linear algebra and some basic group theory. This is a big accomplishment as this material can be quite complex. Fortunately, the paper distills its analysis down to the aspects that will be of most interest to the NeurIPS community.
- **The problem is a natural fit for equivariant neural networks:** The paper makes a convincing case that the problem of computing invariants of topological insulators is one where considerations of symmetry are essential to solving the learning problem. This is significant at a time where equivariance has been abandoned for several tasks in favor of just using more compute (e.g., alphafold). It seems unlikely that a similar approach would work in this setting. This problem would likely be of high interest to the geometric deep learning community.

Weaknesses
- **The architecture section could be better explained:** It is possible that it is just because this reviewer has not worked with gauge equivariant neural networks before, but he found Section 4.1 which describes different layers in GEBLNet and GEConvNet to be a little unmotivated. It would improve the paper if more time was spent describing each layer at a high-level (e.g., what does this function do, what properties does it give the network, etc.). That being said, the reviewer has seen far more impenetrable descriptions in geometric deep learning papers at past NeurIPS.
- **Evidence why GEBLNet works but other architectures do not :** Both the determinant example in Section 3.2 and the DeepSpec example in 3.3 are a nice opportunity to explore what makes other approaches fail. While this is speculated on in these sections, it would make the paper more compelling if deeper experimental evidence was provided to support these claims. As it is, most analysis seems to be based purely on performance.

Nitpicks
- Line 87: ‘…seen an explosive development…’ $\mapsto$ ‘…seen explosive development…’
- Figure 1: It would be interesting to see this plot but with addition curves representing different architecture variations (e.g., values of $R$).

---

> ### Author Rebuttal · Authors · 2025-07-30
>
> We would like to thank the reviewer for providing such a thorough and positive evaluation of our submission. In the following rebuttal, we will try to answer your questions and clear up potential misunderstandings.
>
> ### Weaknesses
>
> > **The architecture section could be better explained:**  It is possible that it is just because this reviewer has not worked with gauge equivariant neural networks before, but he found Section 4.1 which describes different layers in GEBLNet and GEConvNet to be a little unmotivated. It would improve the paper if more time was spent describing each layer at a high-level (e.g., what does this function do, what properties does it give the network, etc.). That being said, the reviewer has seen far more impenetrable descriptions in geometric deep learning papers at past NeurIPS.
> >
>
> We acknowledge the fact that the motivation for different layers were not clearly stated. We will clarify this in the revised version of the paper.
>
> In short, there are three major layers in GEBLNet and GEConvNet: GEBL, GEConv, and GEAct. The samples can be viewed as matrix-valued multichannel “images”, and the operations of the layers are analogous to standard image processing operations, but in a gauge group setting.
>
> GEBL calculates pixel-wise matrix multiplications among channels, yielding a second order polynomial at each site (pixel).  This captures higher-order features for later processing when the matrix-valued samples are converted to scalar-valued outputs.
>
> GEConv serves a similar purpose to the traditional convolution layers where it extracts features among neighborhoods of each pixel with linear combinations. Alternatively from a physics viewpoint, the matrix channel represents an integral of a matrix valued function along a small contour, and GEConv layers combine these to yield integrals along larger curves.
>
> GEAct is a non-linearity that is invariant under gauge transformations.
>
> > • **Evidence why GEBLNet works but other architectures do not :** Both the determinant example in Section 3.2 and the DeepSpec example in 3.3 are a nice opportunity to explore what makes other approaches fail. While this is speculated on in these sections, it would make the paper more compelling if deeper experimental evidence was provided to support these claims. As it is, most analysis seems to be based purely on performance.
> >
>
> We agree that it would be interesting to further investigate the failure modes of other architectures for these examples. For the determinant task, the goal was simply to show that higher order terms (i.e. at least bilinear) are sufficient to predict the determinant of matrices with high rank, whereas standard linear MLPs do not suffice.
>
> For the DeepSpec example, it would be very interesting to better understand why equivariance provides a better model prior even though the task output is invariant. In analogy to equivariant tasks sometimes benefitting from approximate model equivariance when traversing the loss lanscape, it might be the case that our equivariant model provides the needed flexibility in terms of training dynamics. It could also be the case that it is beneficial to perform intermediate computations on equivariant quantities rather than invariant ones. We hope that GEBLNet paves the way for future work into these questions.
>
> ### Nitpicks
>
> > Line 87: ‘…seen an explosive development…’ ‘…seen explosive development…’
> >
>
> Thank you for pointing out this typo.
>
> > Figure 1: It would be interesting to see this plot but with addition curves representing different architecture variations (e.g., values of $R$).
> >
>
> Thank you for this suggestion. We have generated the corresponding plot and will include it in a revised version of the manuscript, however, due to the rebuttal formatting rules, we cannot include it here. The general trend we observe is that models whose total order is lower than the order of terms in the determinant fail to learn the determinant. Additionally, there seems to be an exponential increase in error with matrix size, even for sufficiently large models.
>
> ### Questions
>
> > • Line 154: The term ‘flux data’ appears here without a definition. It is later referenced several times. What does this mean?
> >
>
> Thank you for pointing out this oversight. With the fluxes / flux data, we just mean the Wilson loops $W_k$.
>
> > • Reviewer may have missed it, but what is the point of introducing GEConvNet it is not perform particularly well in any of the instances?
> >
>
> When Favoni et al. (2022) introduced their gauge equivariant models, the GEConv layers were a central building block. Therefore, it is a natural baseline to consider whether these convolutional layers would also enhance performance in our setting. A negative answer suggests a drawback when redundant neighboring information is introduced, which offers valuable insights for later studies.

---

> > ### Comment · Reviewer_FXTi · 2025-08-03
> > **Reply to authors**
> >
> > We thank the authors for answering all of our questions. It is an interesting paper. We will maintain our score.

---

> > > ### Author Response · Authors · 2025-08-06
> > >
> > > Thank you very much for your feedback on our rebuttal and the positive evaluation.
> > >
> > > If you have further questions or comments, we are very happy to provide additional clarifications.

---

### Decision · Program_Chairs · 2025-09-17

**Decision:**

Accept (poster)

**Comment:**

This paper introduces GEBLNet, a gauge-equivariant neural network architecture for learning in the presence of local gauge symmetries. Contributions include a gauge-equivariant normalization layer that stabilizes training, a universal approximation theorem showing the model can approximate any continuous gauge-invariant function, and extensive empirical validation. The model is applied to the prediction of Chern numbers from synthetic lattice gauge configurations, and a notable achievement is that it successfully learns non-trivial and higher Chern numbers for the first time, even when trained only on trivial cases. The authors also extend experiments to 4D Chern numbers and show generalization across grid sizes, with only a moderate linear decrease in accuracy attributed to error accumulation.

Reviewers praised the clarity and accessibility of the paper. The architectural advances are nontrivial, and the demonstration that a gauge-equivariant network can capture global, integer-valued topological invariants represents an important proof-of-concept. Limitations are that experiments are restricted to synthetic, clean data, without tests on DFT/Wannier-derived Hamiltonians, and that the task itself is solvable by analytic methods. A natural next step would be to validate on realistic Hamiltonians, which would strengthen the link to practical physics workflows. However, the main point of this work is to demonstrate that such networks can learn these types of topological quantities at all, rather than to propose a practical replacement for established computational methods.

Overall: While the immediate practical impact is limited, this work makes a meaningful methodological contribution to equivariant ML. Showing that gauge-equivariant networks can stably learn higher Chern numbers is a first, and the techniques developed here should be broadly applicable beyond this specific application.